# Mental Disabilities Increase the Risk of Respiratory Infection-Related Healthcare Utilization

**DOI:** 10.3390/ijerph16203845

**Published:** 2019-10-11

**Authors:** Chen-Hung Chiang, Ming-Che Tsai, Yee-Yung Ng, Shiao-Chi Wu

**Affiliations:** 1Institute of Health and Welfare Policy, School of Medicine, National Yang-Ming University, Taipei 11221, Taiwand40112003@ym.edu.tw (M.-C.T.); 2Department of Gerontology and Health Care Management, Chang Gung University of Science and Technology, Taoyuan City 33303, Taiwan; 3Department of Medicine, School of Medicine, Fu Jen Catholic University, New Taipei City 24205, Taiwan

**Keywords:** mental disability, dental disease, healthcare utilization, respiratory infections, retrospective study, two-part model

## Abstract

Patients with chronic mental illness are highly vulnerable to chronic respiratory problems. We examined the influence of mental disability on respiratory infection-related utilization risk in individuals with and without mental disabilities (MDs). A population-based, retrospective cohort design and two-part model were used to analyze respiratory infection-related utilization in individuals with MDs (MD group) and a matched reference group. The respiratory infection-related utilization rate in one year was lower in the MD group (53.8%) than in the reference group (56.6%). The odds ratios (ORs) were significantly higher among individuals with profound MDs (aOR = 1.10; 95% CI: 1.07–1.14) and those with a history of dental cavities (aOR = 1.16; 95% CI: 1.13–1.19) or periodontal disease (aOR = 1.22; 95% CI: 1.19–1.26) after controlling for covariables. The average number of visits was higher in the MD group (5.3) than in the reference group (4.0). The respiratory infection-related utilization rate and average number of visits were significantly higher in the mild, moderate and severe disabled groups with a history of periodontal disease, respectively, than that of the reference group. In conclusion, healthcare authorities must develop an incentive program to prevent respiratory infections among individuals with MDs.

## 1. Introduction

Although lower respiratory tract infections (LRTIs) are largely preventable [1], respiratory infections such as bacterial pneumonia and bronchitis are common. LRTIs are one of the major causes of mortality worldwide in individuals of all ages, and also the second-leading factor resulting in an increase in disability-adjusted life years (DALYs) [2]. The previous literatures from small sample size showed that patients with mental health disorders were associated with chronic lung disease [3,4]. Filik and colleagues [5] reported a higher rate of lung function impairment in individuals with schizophrenia compared to that those in a national sample in the United Kingdom. Whether the patients with mental disability associated with respiratory infection and respiratory infection-related healthcare utilization is worth to be investigated in Taiwan. Taiwan’s National Health Insurance (NHI) system is a nationwide program that can provide healthcare services for individuals with less of a financial barrier. Universal health coverage is recognized as an avenue for improving equity in health [6]. This nationwide population-based study was conducted to investigate and compare (i) respiratory infection status, and (ii) respiratory infection-related healthcare utilization between individuals with and without mental disabilities. The hypothesis of this study was that the respiratory infection-related utilization rate was increased along with the severity of the mental disability. 

## 2. Materials and Methods 

### 2.1. Study Data and Design

Taiwan’s NHI program currently includes >99.7% of the population of Taiwan and provides universal, compulsory, and low copayment health services that minimize the economic barrier to medical care for inpatient, outpatient, prescription, and various other healthcare services. The NHI database (NHID) contains detailed records of outpatient visits, hospital admissions and emergency department visits, including diagnosis, procedures, medication, provider information, and expenditure. These databases are encrypted for de-identification and have been monitored for completeness and accuracy by Taiwan’s Ministry of Health and Welfare. A population-based, retrospective cohort design was used to analyze the digital claim data between 2013 and 2015 of the NHID, and information relating to disabled individuals (Disabled Population Profile) in the 2014 Taiwan’s Ministry of Health and Welfare Database. All data are stored in the Health and Welfare Data Science Center and only analyzed results can be obtained. This secondary data analysis study was performed following the rules of the Declaration of Helsinki of 1975. This study was approved by the Institutional Review Board of National Yang-Ming University in Taiwan (approval no. YM105043E-3).

### 2.2. Study Subjects

The two study groups involved in the present study were selected from the Disabled Population Profile and NHID, comprising an MD group and a reference group. The study participants encompassed disabled individuals with chronic mental disability of all ages between 1 January 2014, and 31 December 2014, from the Disabled Population Profile. A total of 120,504 individuals in the MD group were eligible for inclusion. Exclusion criteria for the MD group were individuals who died between 1 January 2015, and 31 December 2015 (*n* = 1580) and those with missing data (*n* = 1050). Following exclusion, a total of 117,874 cases were included in the analysis. The reference group was randomly selected from the non-MD group (*n* = 21,853,264) using a 1:2 match with the MD group based on age and sex. Following matching, the MD group (*n* = 117,031) and reference group (*n* = 234,062) were included in the analysis. All study participants were followed up for one year after enrollment. Both groups were followed for one year to investigate healthcare utilization related to respiratory infections.

The dependent variables for this study were the probability of any use and the number of respiratory infection-related healthcare visits for one year. Respiratory infection-related utilization was defined as diagnosed with any of the following respiratory diseases in at least one physician visit or hospitalization in one year: acute nasopharyngitis, acute pharyngitis, acute laryngitis and tracheitis, acute upper respiratory infections, acute bronchitis, pneumonia, influenza and/or bronchitis not specified as acute or chronic (ICD-9-CM codes 460, 462, 464–466, 480–486, 487, and 490). The independent variable was mental disability. The severity of the mental disability was categorized into mild, moderate, severe, and profound by physician according to ICF (International Classification of Functioning, Disability and Health) criteria. 

The relevant covariates included the following: (1) demographic characteristics, such as gender and age; (2) economic status, such as premium-based monthly salary, and community resources, such as the degree of urbanization in the subject’s area of residence (levels 1–7 for highly urbanized cities and townships, moderately urbanized cities and townships, emerging cities and townships, average cities and townships, aging cities and townships, agricultural cities and townships, and remote cities and townships, respectively); (3) health condition, such as dental diseases and chronic disease comorbidity. The history of dental diseases comprised of a history of no dental disease or dental disease (dental cavities and periodontal disease) during the previous year. In this study, only patients who had at least two ambulatory visits or one hospitalization prior to the index date (31 December 2014) with a diagnosis of dental cavities (ICD9-CM codes 521.0) or periodontitis (ICD9-CM codes 523.3–5) were identified as patients with a history of dental cavities or periodontitis. The chronic disease comorbidity was based on the disease score calculated by the D-M Charlson comorbidity index (CCI) definition [7], including CCI = 0 (reference group), CCI = 1 and CCI ≥ 2. 

### 2.3. Statistical Analysis

Programming and data analyses were performed using SAS statistical software, version 9.4 (SAS Institute Inc., Cary, NC, USA). Descriptive statistics are represented by the numbers of cases, percentages, and means with standard deviation. The frequency and percentage of respiratory infection-related utilization were then analyzed. The Chi-square test and t-test were used to compare the proportions and means between the two study groups. A two-part model was used to analyze respiratory infection-related utilization. The first part was a binary equation to model the probability of any respiratory infection-related utilization; for the second part, a generalized linear model was used to explain the number of respiratory infection-related healthcare utilizations in cases with at least one physician visit or hospitalization in one year. Conditional logistic regression and generalized linear models were used to analyze related influential factors associated with respiratory infection-related healthcare utilization in patients with MDs, and the significance level was *p* < 0.05. Multicollinearity was also analyzed via the variance inflation factor (VIF) using regression analysis. As the VIF of each coefficient was <5, it was considered that the effect of correlation among the independent variables was not sufficient to distort the estimation. 

## 3. Results

### 3.1. Analysis of the Basic Characteristics of Individuals with Mental Disabilities and Reference Groups

A total of 117,031 individuals with MDs and 234,062 individuals without MDs were recruited for the study. The matched data showed that there was no significant difference in gender, age group or history of periodontal disease between the MD and reference groups (Table 1). The level of premium-based monthly salary in the MD group was significantly lower than reference group (6.7% versus 10.2% in monthly salary below US$760, 85.9% versus 58.4% in the US$760–1160 category, and 7.4% versus 31.4% above US$1160 category, *p* < 0.0001). Out of the study participants of the MD group, 12.7% had a history of dental cavities. 

### 3.2. Analysis of Respiration Infection in Individuals of the MD and Reference Groups

This study found that the utilization of respiratory infections in individuals with MDs was associated with various factors, including gender, age, premium-based monthly salary, urbanization of residential area, CCI, disability severity, history of dental cavities and history of periodontal disease (Table 2). The overall probability of respiratory infections for one year in the MD group (53.8%) was lower than that in the reference group (56.6%). Across the three categories of MD groups, the probability was significantly higher in women, aged <20 years old, and lower monthly salary (<US$760) among individuals with different severities of disability (*p* < 0.001). The probability increased with the level of urbanization of residential location. The probability in individuals with profound MDs was 59.8% for those residing in locations with a low level of urbanization and 61.1% for those residing in locations with high-level urbanization. The probability increased with an increase in the CCI and was highest in the severe (CCI ≥ 2) group (*p* < 0.001). The utilization was significantly higher for those in the MD group with a history of dental cavities (62.8%) compared with that in those without such a history (52.5%). Similarly, among those in the MD group, the probability was significantly higher for those with history of periodontal disease (63.6%) compared with that in those without such a history (51.6%). (Table 2) 

The adjusted model revealed that the probability of respiration infection increased with the progression of disability severity. The probability of respiration infection was 1.10 times (95% CI: 1.07–1.14) higher in the profoundly MD group, but 0.66 to 0.80 times lower in the mild, moderate and severe MD group than the reference group (*p* < 0.001) (Table 3). The probability of respiration infection was 0.87 times (95% CI: 0.85–0.90) lower in the medium income group (US$760–1160) than in the low-income group (lower than US$760). There was a 0.92 times (95% CI: 0.89–0.94) and 0.97 times (95% CI: 0.95–1.0) lower probability of respiration infection in low and medium urbanization of residence areas than in high urbanization areas. The probability of respiration infection in patients with CCI = 1 or CCI ≥ 2 were 1.72 times (95% CI: 1.68–1.76) and 1.85 times (95% CI: 1.80–1.91) higher than patients with CCI = 0, respectively. The probability of respiration infection was 1.16 times (95% CI: 1.13–1.19) and 1.22 times (95% CI: 1.19–1.26) higher in those with a history of dental cavities and periodontal disease than those without such histories, respectively. The regression analysis model was used with an interaction term between mental disability and dental cavities/periodontal disease history variables to examine the effect of mental disability and dental diseases/periodontal disease on the probability of respiration infection. There was no significant difference regarding the interaction between mental disability and dental cavities history. The probability of respiration infection was 1.29 times (95% CI: 1.17–1.42) and 1.26 times (95% CI: 1.19–1.33) higher in the people with mild, moderate, and severe disability in MD group with a history of periodontal disease, respectively, than that of the reference group without such a history (*p* < 0.001) (Table 3).

### 3.3. Analysis of the Visits for Respiratory Infection-Related Utilization in Individuals of the MD and Reference Groups 

We analyzed the number of visits for respiratory infection disease only in those with utilization. As shown in Table 4, the average number of visits in the MD group (5.3) was significantly higher than that in the reference group (4.0). The average number of visits was significantly higher in women, >20-years-old, low and medium month salary, high urbanization area, high CCI, history of dental cavities, and history of periodontal disease than men (*p* < 0.001), <20-years-old (*p* < 0.001), high month salary (*p* < 0.001), lower urbanization area (*p* < 0.001), low CCI (*p* < 0.001), no history of dental cavities (*p* < 0.001), and no history of periodontal disease (*p* < 0.001), respectively. 

After controlling for potential confounding covariates, the generalized linear regression model showed that the average number of visits related to respiratory infection in the MD group was significantly higher than the reference group (*p* < 0.001) (Table 3). The average number of visits in the profoundly disabled group was 0.98 higher than that in the reference group (*p* < 0.001). Additionally, the average number of visits in the high-income group and low level of urbanization area were 0.22 and 0.24 significantly lower than that in the lowest income group (*p* < 0.001) and high urbanization area (*p* < 0.001), respectively. The average number of visits in those with mild (CCI = 1) and severe (CCI ≥ 2) CCI were 0.98 and 1.52 significantly higher than the people without CCI = 0 (*p* < 0.001). The average number of visits in people with a history of dental cavities and periodontal disease were 0.22 and 0.14 significantly higher than those without such a history (*p* < 0.001), respectively. The average number of visits were 0.59 and 0.34 significantly higher in the mild, moderate, and severely disabled groups with a history of periodontal disease, respectively than that in the reference group without such a history (*p* < 0.001). 

## 4. Discussion

This study yielded three noteworthy results. Firstly, we identified significant differences between individuals with and without MDs regarding respiratory infection-related healthcare utilization. The main results of the study indicate that, compared with the general population, individuals with MDs were less likely to receive medical care due to respiratory infections within one year. However, the utilization of respiratory infections among individuals with profound MDs was significantly higher than that in the reference group (aOR = 1.10; 95% CI 1.07–1.14). Universal coverage of healthcare aims to secure access to appropriate healthcare for all at an affordable cost. Since 1995, Taiwan’s NHI system has provided an equal package of benefits, including outpatient, inpatient, dental and pharmaceutical services for all. Reduced copayment and other welfare programs are available to the elderly and vulnerable groups. Access to primary healthcare (PHC) is a fundamental human right and central to the performance of healthcare systems, however, persons with disabilities (PWDs) generally experience greater barriers in accessing PHC than those in the general population. According to one previous study, individuals with psychotic disorders (OR = 0.55, 95% CI 0.44–0.69) and bipolar disorder (OR = 0.74, 95% CI 0.56–0.98) had significantly reduced odds of having access to a primary healthcare physician compared with individuals without mental disability. Individuals with psychotic disorders, bipolar disorder or major depressive disorder have higher odds of reporting difficulties in accessing care (OR = 2.5–7.0) [8]. PWDs are unable to access PHC due to obstacles that include the interplay of four major factors of availability, acceptability, geography and affordability [9]. There are several possible explanations for why individuals with schizophrenia receive less healthcare. Those with schizophrenia may have problems describing their medical symptoms to primary care physicians. Physicians may also be uncomfortable treating people with schizophrenia [10], possibly reflecting the stigmatization of patients with this disorder. Similarly, psychiatrists may not feel comfortable providing primary and preventive healthcare to their patients [11]. 

Secondly, after controlling for potential confounding covariates, the average respiratory infection-related healthcare utilization visits in individuals with MDs was significantly higher compared with that in the reference group and increased with increased disability severity. Previous studies have demonstrated a higher prevalence of respiratory disease [12], respiratory symptoms and impaired lung function [5] in individuals with serious mental illness compared with that in the general population. Patients with schizophrenia had more than twice the odds of suffering from asthma (OR, 2.43; 95% CI, 1.37–4.32), more than three times the odds of suffering from chronic bronchitis (OR, 3.69; 95% CI, 2.00–6.81), and nine times the odds of suffering from emphysema (OR, 9.14; 95% CI, 4.12–20.31). Patients with affective disorder also had significantly elevated rates of the following comorbid conditions: asthma (OR, 2.56; 95% CI, 1.50–4.38), chronic bronchitis (OR, 4.57; 95% CI, 2.74–7.61) and emphysema (OR, 4.19; 95% CI, 1.69–10.36) [13]. Most associations between 16, mental disability and the subsequent onset or diagnosis of chronic lung disease were statistically significant, with ORs (95% CIs) ranging from 1.6 (1.1–2.3) for posttraumatic stress disorder to 3.0 (2.0–4.7) for intermittent explosive disorder [3]. In patients with pneumonia, those with schizophrenia were independently associated with a 1.81 times greater risk of intensive care unit admission (95% CI 1.37–2.40), a 1.37 times greater risk of acute respiratory failure (95% CI 1.08–1.88) and a 1.34 times greater risk of mechanical ventilation (95% CI 1.04–1.92) after adjusting for the characteristics of patients, physicians, hospitals, and potential clustering effects. Significant barriers to prompt and appropriate medical care for pneumonia persist for patients with schizophrenia [14]. There is also evidence that patients with schizophrenia have less access to healthcare [15,16,17]. Once hospitalized, adverse events during and after medical interventions occur more frequently than in patients without schizophrenia [18]. Patients with psychiatric disorders, particularly those with severe mental illness, have high rates of undetected and untreated medical problems and substantially elevated mortality rates due to medical illness [12,19]. Additionally, a problem of many healthcare systems is that psychiatry is not integrated into a general medical setting. Therefore, patients with psychiatric problems do not have adequate access to medical treatment. In many psychiatric centers, there may be a lack of resources for performing appropriate laboratory examinations and treatment interventions.

Finally, this study revealed a higher association between dental diseases and respiratory infection-related healthcare utilization. The respiratory infection-related healthcare utilization rates and the average number of visits were significantly higher in the mild, moderate, and severely disabled groups with a history of periodontal disease, respectively, than in the reference group. Oral bacteria, poor oral hygiene, and periodontitis influence the incidence of pulmonary infections. The oral colonization by potential respiratory pathogens, possibly fostered by periodontitis, and possibly by bacteria specific to the oral cavity or to periodontal diseases contribute to pulmonary infections [20]. The amount of calculus and colonization of the tongue with respiratory pathogens are risk factors for the development of pneumonia. Oral hygiene measures to remove the tongue biofilm and calculus may reduce its development [21]. Oral bacteria may contribute to the etiology of respiratory diseases. Four possible mechanisms, including oral bacteria and salivary enzymes to explain the biological plausibility of an association between oral conditions and respiratory infections, have been proposed [22]. The oral health status among individuals with MDs is poorer than that in the general population. Poor oral health has been reported among various psychiatric populations. The oral health status of a group of Chinese psychiatric inpatients in a long-term rehabilitation facility was poor. Bleeding on probing, calculus, and shallow and deep pockets were found in 7.1%, 71.8%, 72.9%, and 28.2% of patients, respectively. Dental caries were found in 75.3% of dental patients [23]. The ORs for caries were significantly higher among adults with intellectual disability (ID) and in individuals with co-occurring developmental disorders (DDS) and increased with the level of ID. Those with DDS were associated with a 1.6 times higher likelihood of having untreated decay, and institutionalization was associated with a 2.4 times higher likelihood of having untreated decay. Institutionalization and co-occurring disabilities have been found to be significantly associated with a higher probability of developing gingivitis [24]. The prevalence and incidence of adverse oral health outcomes in adults with serious mental illnesses are controversially reported to be higher or marginally similar to those in the general population [25]. Therefore, individuals with MDs and a history of periodontal disease should be given more attention regarding possible respiratory infections.

This study has several strengths. Firstly, using a nationwide population-based dataset, the number of subjects provides ample statistical power to detect differences between groups after adjusting for confounding variables. A comparison of respiratory infection-related healthcare utilization outcomes with the general population was also possible. Secondly, the study participants encompassed different severities of mental disability for chronic mental disability. The severity of the MDs was credible through professional diagnosis and identification by physicians. The resulting outcomes show a statistically significant dose-response relationship between the severity of the mental disability and respiratory infection-related healthcare utilization. Finally, the dental diseases in question comprised the presence or absence of a history of dental cavities and periodontal disease in the previous year. A regression analysis model with an interaction term between mental disability and periodontal disease history variables was used to examine the effect of mental disability and dental diseases on respiratory infection-related healthcare utilization. The presence of interactions can have important implications for the interpretation of statistical models.

Given the nature of secondary data analysis, our study has some limitations. First, we could not exclude the patients with mental disorders in the reference group if physicians did not make the diagnosis codes of mental disability in their claim form. However, physicians try hard to record all the diagnosis of their patients in claim form in order to reimburse payment from NHI. On the other hand, the reference group (234,062) was coming from non-MD group (*n*= 21,853,264). Therefore, the non-excluded cases of mental disability in the reference group might be very limited and not influence the study result. Second, compared to the reference group, people with mental disabilities are often excluded as candidates for organ transplantation in clinical practice. Therefore, respiratory infection-related utilization rate and average number of visits may influence the reference group more than the mental disability group. The results of a significant higher respiratory infection-related utilization rate and average number of visits in mental disability group might not be changed. Third, we were unable to obtain a history of alcohol intake and smoking data for all subjects to discuss those effect of respiratory infection. Fourth, we were able to count the number of respiratory infection utilization only in patients who were treated. For patients with respiratory tract infections who do not receive medical treatment, it is impossible to calculate utilization. Because of access to affordable low health insurance copayment and health care providers in Taiwan, the economic barrier of mental disability people is limited. Therefore, the underestimation of the association between mental disability and respiratory may be limited. Fifth, we were unable to record the oral health behavior of all subjects or dental care utilization data for individuals who did not receive care due to dental diseases. The oral health status among individuals with MDs was poorer than that of the general population. As patients with severe mental illness visit dentists less frequently than those in the general population in Taiwan [26], our data may underestimate the association between dental diseases and respiratory infection-related healthcare utilization. 

## 5. Conclusions

In summary, there were significant differences between individuals with and without MDs in respiratory infection-related healthcare utilization. Individuals with MDs received less medical care than those in the general population in Taiwan. Once they had received medical care, individuals with MDs used services more frequently than those without MDs. The respiratory infection-related utilization rate and average number of visits were significantly higher in the mild, moderate and severe disabled groups with a history of periodontal disease, respectively, than in the reference group. Changing the care-seeking behavior of individuals with MDs in the short-term would be difficult; therefore, healthcare authorities should organize health systems to facilitate meeting the medical needs of individuals with MDs. Healthcare providers, including psychiatrists, dentists, nursing staff in institutions, and caregivers, should encourage those with MDs to regularly visit the dentist and strengthen their oral health literacy, so as to improve the oral health of these vulnerable groups. There must be greater investment in early intervention and the development of service plans that will deliver accessible provision over time, which may reduce the risk of respiratory infections for the mentally disabled. Appropriate, quality healthcare should be equitably and accessibly delivered across the country and to all groups of individuals with mental disabilities.

## Figures and Tables

**Table 1 ijerph-16-03845-t001:** Characteristics of individuals in the MD and reference groups before and after matching.

Variables	Pre-Matched		*p* Value	Matched		*p* Value
	Mental Disabilities	Reference		Mental Disabilities	Reference	
	(*n* = 117,874)	(*n* = 21,853,264)		(*n* = 117,031)	(*n* = 234,062)	
	%	%		%	%	
Gender			0.0687			>0.9999
Male	49.0	49.2		49.0	49.0	
Female	51.0	50.8		51.0	51.0	
Age group (years)			<0.0001			>0.9999
<20	0.5	21.4		0.5	0.5	
20–44	34.5	38.9		34.6	34.6	
45–64	53.6	29.0		53.5	53.5	
≥65	11.4	10.8		11.4	11.4	
Premium-based monthly salary (US$)			<0.0001			<0.0001
<760 (Q1)	6.7	13.2		6.7	10.2	
760–1160	85.9	58.4		85.9	58.4	
>1160 (Q3)	7.4	28.4		7.4	31.4	
Urbanization of residence area			<0.0001			<0.0001
Low	23.3	28.4		23.3	29.0	
Medium	52.5	53.6		52.5	53.5	
High	24.2	18.0		24.2	17.6	
Charlson comorbidity index (CCI)			<0.0001			<0.0001
No (CCI = 0)	62.6	79.4		62.6	73.9	
Mild (CCI = 1)	20.4	12.7		20.4	15.3	
Severe (CCI ≥ 2)	17.0	7.9		17.1	10.7	
Severity of disability			<0.0001			<0.0001
Non		100.0			100.0	
Mild	2.1			2.1		
Moderate	17.5			17.5		
Severe	54.2			54.2		
Profound	26.2			26.3		
Dental cavities history			<0.0001			<0.0001
Yes	12.7	14.1		12.7	11.8	
No	87.3	85.9		87.3	88.2	
Periodontal disease history			<0.0001			0.8634
Yes	18.5	15.8		18.5	18.5	
No	81.5	84.2		81.5	81.5	

Note. MD: Mental disabilities, Distribution among groups was analyzed by the Chi-square test.

**Table 2 ijerph-16-03845-t002:** Rates of respiratory infection-related healthcare utilization among individuals in the MD and reference groups.

Variables	MD Group												Reference Group		
	Mild, Moderate			Severe			Profound			Total					
	*n*	%	*p*	*n*	%	*p*	*n*	%	*p*	*n*	%	*p*	*n*	%	*p*
Total	22,839	47.1		63,447	52.7		30,745	61.1		117,031	53.8		234,062	56.6	
Gender			<0.0001			<0.0001			<0.0001			<0.0001			<0.0001
Male	12,030	45.5		32,046	48.5		13,271	55.3		57,347	49.4		114,694	51.7	
Female	10,809	48.9		31,401	56.9		17,474	65.5		59,684	58.0		119,368	61.3	
Age group (years)			<0.0001			<0.0001			<0.0001			<0.0001			<0.0001
<20	54	61.1		181	71.8		392	75.5		627	73.2		1254	69.2	
20–44	5325	49.0		22,123	55.3		13,042	63.2		40,490	57.0		80,980	57.7	
45–64	13,102	45.1		34,822	51.1		14,701	59.5		62,625	51.8		125,250	55.0	
≥65	4358	50.6		6321	51.7		2610	57.5		13,289	52.5		26,578	60.6	
Premium-based monthly salary (US$)			<0.0001			<0.0001			<0.0001			<0.0001			<0.0001
<760 (Q1)	654	54.4		3819	58.9		3377	65.2		7850	61.2		23,951	58.6	
760–1160	21,056	46.7		55,452	51.9		23,994	60.4		100,502	52.8		136,675	55.3	
>1160 (Q3)	1129	50.5		4176	57.1		3374	61.9		8679	58.1		73,436	58.4	
Urbanization of residence area			0.007			<0.0001			0.016			<0.0001			0.02
High	6809	48.2		15,654	51.1		5825	61.1		28,288	52.5		41,124	57.1	
Medium	11,610	47.2		33,309	53.4		16,519	61.7		61,438	54.5		125,123	56.8	
Low	4420	45.2		14,484	52.5		8401	59.8		27,305	53.6		67,815	56.0	
Charlson comorbidity index (CCI)			<0.0001			<0.0001			<0.0001			<0.0001			<0.0001
No (CCI = 0)	14,016	40.7		39,706	46.9		19,481	57.1		73,203	48.4		173,027	53.5	
Mild (CCI = 1)	4699	55.2		12,988	60.1		6166	67.2		23,853	61.0		35,880	65.8	
Severe (CCI ≥ 2)	4124	59.7		10,753	64.7		5098	68.8		19,975	64.7		25,155	64.8	
Dental cavities history			<0.0001			<0.0001			<0.0001			<0.0001			<0.0001
No	20,580	46.2		55,340	51.2		26,214	60.0		102,134	52.5		206,341	55.9	
Yes	2259	55.3		8107	62.5		4531	67.0		14,897	62.8		27,721	62.3	
Periodontal disease history			<.0001			<0.0001			<0.0001			<0.0001			<0.0001
No	19,422	45.2		51,692	50.3		24,283	59.4		95,397	51.6		190,738	55.4	
Yes	3417	57.6		11,755	63.2		6462	67.3		21,634	63.6		43,324	62.0	

**Table 3 ijerph-16-03845-t003:** Influential factors for respiratory infections utilization among individuals in the mental disabilities and reference groups.

Variables	Respiratory Infections^1^						Visits of Respiratory Infections Utilization (Only Users)^2^					
	Crude OR	95% CI	*P*	Adj OR	95% CI	*P*	Crudeβ	SE	*P*	Adj β	SE	*P*
Intercept										3.85	0.05	<0.0001
Severity of mental disability (Reference: non)												
Mild, Moderate	0.68	0.66–0.71	<0.0001	0.66	0.63–0.68	<0.0001	1.19	0.06	<0.0001	0.77	0.06	<0.0001
Severe	0.86	0.84–0.88	<0.0001	0.80	0.78–0.82	<0.0001	1.34	0.03	<0.0001	0.99	0.04	<0.0001
Profound	1.17	1.14–1.20	<0.0001	1.10	1.07–1.14	<0.0001	1.24	0.04	<0.0001	0.98	0.05	<0.0001
Premium-based monthly salary (Reference: <760 US$)												
760–1160	0.85	0.83–0.88	<0.0001	0.87	0.85–0.90	<0.0001	0.32	0.04	<0.0001	0.01	0.04	0.8741
>1160	1.04	1.01–1.08	0.0159	1.02	0.99–1.06	0.166	−0.35	0.05	<0.0001	−0.22	0.05	<0.0001
Urbanization of residence area (Reference: High)												
Medium	1.02	1.00–1.05	0.0421	0.97	0.95–1.00	0.018	−0.31	0.03	<0.0001	−0.17	0.03	<0.0001
Low	0.99	0.97–1.01	0.3575	0.92	0.89–0.94	<0.0001	−0.44	0.04	<0.0001	−0.24	0.04	<0.0001
Charlson comorbidity index (CCI) (Reference: CCI = 0)												
Mild (CCI = 1)	1.69	1.65–1.73	<0.0001	1.72	1.68–1.76	<0.0001	1.11	0.03	<0.0001	0.98	0.03	<0.0001
Severe (CCI ≥ 2)	1.80	1.75–1.85	<0.0001	1.85	1.80–1.90	<0.0001	1.73	0.04	<0.0001	1.52	0.04	<0.0001
Dental cavities history (Reference: No)												
Yes	1.34	1.31–1.37	<0.0001	1.16	1.13–1.19	<0.0001	0.40	0.04	<0.0001	0.22	0.04	<0.0001
Periodontal disease history (Reference: No)												
Yes	1.41	1.38–1.44	<0.0001	1.22	1.19–1.26	<0.0001	0.40	0.03	<0.0001	0.14	0.04	0.0004
MD*PD (Reference: No)												
Mild, Moderate*PD	-	-	-	1.29	1.17–1.42	<0.0001	-	-	-	0.59	0.14	<0.0001
Severe*PD	-	-	-	1.26	1.19–1.33	<0.0001	-	-	-	0.34	0.08	<0.0001
Profound*PD	-	-	-	1.04	0.96–1.12	0.333	-	-	-	0.27	0.10	0.0069

Note. Only users: all persons with at least one physician visit or hospitalization in one year, 1: conditional logistic model, 2: GLM Model, SE: standard error, 95% CI: 95% confidence intervals, adj OR: adjusted odds ratio, adj β: adjusted β, MD: severity of mental disability, PD: periodontal disease history.

**Table 4 ijerph-16-03845-t004:** Visits of respiratory infections utilization among only use individuals in the MD and reference groups.

Variables	MD Group								Reference Group		*p* Value *
	Mild, Moderate		Severe		Profound		Total				
	(*n* = 10,755)		(*n* = 33,409)		(*n* = 18,777)		(*n* = 62,941)		(*n* = 132,553)		
	Mean	SD	Mean	SD	Mean	SD	Mean	SD	Mean	SD	
Total	5.2	7.11	5.4	7.18	5.3	6.93	5.3	7.09	4.0	4.54	<0.0001
Gender											<0.0001
Male	5.3	7.66	5.2	7.30	4.9	6.97	5.1	7.29	3.8	4.39	
Female	5.1	6.48	5.5	7.07	5.5	6.88	5.5	6.92	4.2	4.65	
Age group (years)											<0.0001
<20	4.6	4.65	4.9	4.91	4.4	4.33	4.5	4.52	3.6	3.98	
20–44	5.5	8.71	5.2	6.88	5.1	6.85	5.2	7.10	3.6	3.81	
45–64	5.2	6.77	5.5	7.47	5.5	7.13	5.5	7.25	4.1	4.71	
≥65	5.0	5.77	5.3	6.69	5.6	6.48	5.2	6.37	4.9	5.54	
Premium-based monthly salary (US$)											<0.0001
<760(Q1)	5.4	7.67	5.3	7.48	5.2	7.50	5.3	7.50	4.0	4.48	
760–1160	5.2	7.14	5.4	7.20	5.3	6.98	5.4	7.13	4.2	4.73	
>1160(Q3)	5.1	6.08	5.2	6.54	5.0	5.82	5.1	6.20	3.8	4.20	
Urbanization of residence area											<0.0001
High	5.7	7.25	5.7	7.68	5.4	7.23	5.6	7.48	4.2	4.65	
Medium	5.1	7.31	5.3	6.82	5.3	6.91	5.2	6.93	4.1	4.59	
Low	4.8	6.19	5.4	7.44	5.2	6.73	5.3	7.04	3.9	4.39	
Charlson comorbidity index (CCI)											<0.0001
No (CCI = 0)	4.7	6.23	4.7	6.16	4.7	6.25	4.7	6.20	3.7	4.02	
Mild (CCI = 1)	5.4	6.50	5.9	7.64	5.8	7.20	5.8	7.32	4.7	5.19	
Severe (CCI ≥ 2)	6.2	9.19	6.6	8.79	6.6	8.27	6.5	8.73	5.1	5.87	
Dental cavities history											<0.0001
No	5.2	6.76	5.3	7.05	5.2	6.96	5.3	6.97	4.0	4.51	
Yes	5.6	9.32	5.9	7.85	5.7	6.73	5.8	7.72	4.3	4.73	
Periodontal disease history											<0.0001
No	5.1	6.52	5.2	6.92	5.2	6.91	5.2	6.85	4.0	4.53	
Yes	5.9	9.27	5.9	8.01	5.7	6.94	5.8	7.89	4.2	4.60	

Note. Only users: all persons with at least one physician visit or hospitalization in one year. *p* value *: comparison between total mental disabilities and the reference group.

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
