# Peer review of "Mental Disabilities Increase the Risk of Respiratory Infection-Related Healthcare Utilization"

_ijerph, 2019, doi:10.3390/ijerph16203845_

Round 1

Reviewer 1 Report

Introduction is clear, but there are some issues to clarify in methods and results
- Line 79: It is not clear in study objectives (there are two) that the researchers are looking an association between dental diseases in individuals with or without mental disorders.
- Lines 102 and 108: In the reference group, people with mental disorders have been excluded? have you considered excluding people with severe immunosuppression, as in solid-organ or bone-marrow transplantation or Human immunodeficiency virus infection/acquired immune deficiency syndrome (HIV/AIDS), or receiving chemotherapy or other immunosuppressive drugs. Also, Do you hace access to medication history, anti-acid treatment? inhaled corticosteroids?
- Line 109: You mentioned that in the analysis you match the two groups based on parameters including age and sex. You imply that there are more parameters?, or only two?
- Line 111-119: You defined infection-related utilization as a persona with any respiratory disease in at least one physician visit or hospitalization in one year, so if a person had only one infection in the year, or ten infections in the year, you consider it the same?
- line 121: what tools do you use to categorize the severity of the mental disorders?
- line 124: What does it mean "index date"
- Results: In general, I think that is not necessary repeat in the text some numerical results listed in tables or graphics.
- Table 1: Can you include information about chronic comorbidities (cardiovascular, respiratory), in addition to the JRC.

Reviewer 2 Report

Line 32. " Respiratory infectious diseases" I suggest respiratory sepsis or respiratory infections.
Line 34. " The treatment of costly" can be omitted. Our daughter should mention the exact cost of treatment.

Line 34. " Substantial public health problem" can be omitted as the arteries mentioned that this is the fifth oral because of mortality and leading because of illness.

Line 37 " LRTI are largely preventable" needs justification in reference

Line 40 and 41 he also been restructured

Line 42 " chronic medical illness" needs to be defined in the study.

Introduction is lengthy and redundant. This can be added in the discussion section. Daughter should only give introduction to the topic of the study rather than discussing it in the section

Line 121 and 122, who did the hospitals calcified mental disability into mild to moderate in profound?

The article is lengthy and has several redundant information. The authors should focus on studying one major endpoint rather than several. The results of the study can be reported in an seperate articles. 

Reviewer 3 Report

This manuscript presents data on two important topics that are usually overlooked: the co-existence of mental and dental illness. 

I provides valuable information about health care utilization.

I recommend the following: 

1) Even though the purpose of the work is clearly stated, the authors have not stated the research questions or the hypothesis. These could guide the readers through the results and the discussion. It would also be helpful if each of the research questions is paired with the data analysis method that was used to answer each of them. 

1) The terms mental disorders and mental disability are use interchangeably. I recommend to use only mental disability, including in the title. The term itself and the explanation of the  measurement need further clarification. It is not clear what the classification of mild, moderate, severe and profound refers to.
